



# A physics-based probabilistic forecasting model for rainfall-induced shallow landslides at regional scale
Shaojie Zhang[a], Luqiang Zhao[b], Ricardo Delgado-Tellez[c], Hongjun Bao[b]
[a]Key Laboratory of Mountain Hazards and Earth Surface Process, Institute of Mountain Hazards and Environment, Chinese Acade-
my of Sciences, Chengdu 610041, China;
[b]Public Meteorological Service Center of CMA, China Meteorological Administration, Beijing 100081, China
[c]Nipe Sagua Baracoa mountain office, Ministry of Science, Technology and Environment of Cuba, Guantanamo, Cuba
*Correspondence to*: L.Q. Zhao (zhaolq@cma.gov.cn)
**Abstract:** Conventional outputs of physics-based landslide forecasting models are presented as deterministic
warnings by calculating the safety factor ($Fs$) of potentially dangerous slopes. However, these models are highly
dependent on variables such as cohesion force and internal friction angle which are affected by high degree of
uncertainty especially at a regional scale, which result in unacceptable uncertainties of $Fs$. Under such circum-
stances, the outputs of physical models are more suitable if presented in the form of landslide probability values.
In order to develop such models, a method to link the uncertainty of soil parameter values with landslide probabil-
ity is devised. This paper proposes the use of Monte Carlo method to quantitatively express uncertainty by as-
signing random values to physical variables inside a defined interval. The inequality $Fs<1$ is tested for each pixel
in $n$ simulations which are integrated in a unique parameter. This parameter links the landslide probability to the
uncertainties of soil mechanical parameters and is used to create a physics-based probabilistic forecasting model
for rainfall-induced shallow landslides. The prediction ability of this model was tested in a case study, in which
simulated forecasting of landslide disasters associated to heavy rainfalls on July 9 of 2013 in the Wenchuan
earthquake region of Sichuan province, China was performed. The proposed model successfully forecasted land-
slides in 159 of the 176 disaster points registered by the geo-environmental monitoring station of Sichuan prov-
ince. Such testing results indicate that the new model can be operated in a high efficient way and show more reli-
able results attributing to its high prediction accuracy. Accordingly, the new model can be potentially packaged
into a forecasting system for shallow landslides providing technological support for the mitigation of these disas-
ters at regional scale.
**Keywords:** Landslide, probabilistic forecasting, infinite slope model, hydrological process simulation
## 1 Introduction
Rainfall-induced shallow landslides are common in many mountainous areas and are considered extremely
dangerous (Varnes, 1978). In despite of the low volume of debris deposits involved in these processes (generally <
1,000 m$^3$), rainfall-induced shallow landslides present high moving speeds (Cruden and Varnes, 1996), evolve
very rapidly, and can propagate even in presence of obstacles (Davide T. and Davide R., 2010). Current regional
landslide forecasting models mainly focuses on shallow landslides. They can be classified in three categories:
statistics-based methods (Caine, 1980; Crosta, 1998; Crosta and Frattini, 2001; Aleotti, 2004; Wei et al., 2004;
Wieczorek and Glade, 2005; Cardinali et al., 2006; Jacob et al., 2006), contributor-factor-based forecasting meth-
ods (Dai and Lee 2003; Wei et al., 2007a; Chang et al. 2008) and physics-based forecasting methods (Montgom-
ery and Dietrich, 1994; Wu and Sidle, 1995; Montgomery et al., 1998; Iverson, 2000; Wilkinson et al., 2002;
Crosta and Frattini, 2003; Salciarini et al., 2006). The physics-based forecasting models have overcome the draw-
back of statistics-based models with respect to excessive dependence on rainfall data. Furthermore, by devising
mechanisms for coupling rainfall with soil surface mechanics using hydrological process simulation (Zhang et al.,
2014a), the physically-based models represent an improvement over the independent treatment of these factors by
contributor-factor-based forecasting models e.g. (Wei et al., 2007a).
The physics-based forecasting model is able to describe the variation rule of hydrological parameters induced
by rainfall infiltration and further explain the failure mechanism of a slope due to the variation of hydrological
parameters. Those characteristics explain the interest of scholars to the physics-based forecasting model and its
implementation at regional scales (Schmidt et al., 2008; Montrasio et al., 2011; Raia et al., 2014). The most com-
mon analysis unit used in physics-based forecasting models is the pixel, used for example in the well-known
TRIGRS model (Baum, et al., 2002, 2008). The safety factor of each pixel within a forecasting region, $F_s$ ($F_s=R/S$:
where $R$ is shear resistance and $S$ is the driving force) is calculated considering rainfall infiltration, pixels are then
identified as unstable ($F_s > 1$) or stable ($F_s < 1$). From these results, landslide warnings are expressed determinis-
tically by labeling each pixel of the forecasting area as either 'landslide occurrence' or 'nonoccurrence'.
However, it must be noted that the underlying physics-based forecasting model requires large number of sur-
face data to be assigned to each pixel before safety factors can be calculated. The physics-based model is sensitive
to the accuracy of such data, especially the soil mechanical parameters (cohesion force and internal friction angle)
that can significantly influence the pixel stability. In general, and specially for large areas, seemingly deterministic
soil mechanical parameters at pixel level used in physical models have different amounts of uncertainty (Schmidt
et al., 2008; Rossi et al., 2013), which thus generate uncertain forecasting results. In this scenario, it is unwise to
give deterministic forecasting results to the public while using the physical model in local forecasting service.
Providing probabilistic landslide forecasting results is the more direct solution to this issue. Currently, several
scholars advance in the development of physics-based probabilistic forecasting models (Schmidt et al., 2008; Raia
et al., 2014). However, the relationship between the landslide probability and the uncertainties in soil mechanical
parameters is not addressed in their models. This effectively renders such probabilistic models actually still in
deterministic mode. For example, in Raia et al. (2014) a series of deterministic forecasting results are generated by
the model during the simulation process from which an experienced forecaster with professional knowledge of
landslides is necessary for picking up the most probable one. Consequently, this approach requires a large number
of calculations, which is unsuitable for operational forecasting of shallow landslides.
This paper focuses on an effective method for linking landslide probability to the uncertain soil mechanical
parameters. It uses Monte Carlo methods to propose a probabilistic forecasting model with a high calculating
efficiency. The proposed model can directly generate probabilistic forecasting results instead of serial of deter-
ministic results, and hence it will be more suitable to operational forecasting of shallow landslides, in special at
the regional scale.
The next section introduces the physics-based probabilistic forecasting for shallow landslides model. Third
section addresses the general aspects of its application to a regional scale shallow landslide forecasting system.
Fourth section describes a case study in which the effectiveness of the proposed model is analyzed in a study case.
Sections five and six discuss the results and states the conclusions of this study respectively.
**2 Probabilistic forecasting for shallow landslides**
**2.1 The Infinite slope model for unsaturated soil slopes using safety factor $F_s$**
There are two mechanisms that trigger failure in slopes subject to rainfall infiltration. They are loss of matrix
suction and increasing of a positive pore water pressure (Li et al., 2013). In southwestern China, precipitation is
rich in summer due to monsoon conditions from both Pacific and India Ocean (Wei et al., 2006). Before of the
raining season slopes in this area are generally unsaturated during the relatively dry seasons. Almost all landslide
disasters in southwestern China occur during the rainy season when the matrix suction of topsoil's suddenly de-
creases due to monsoon heavy rains. Consequently, this research focuses on the stability analysis of unsaturated
soil mass.
During the evolution process from stability to failure driven by rainfall infiltration, the rapid loss of suction
due to the increasing soil water content is the key triggering factor for shallow landslides. The safety factor $F_s$ is
used to evaluate the stability of slopes under the action of rainfall infiltration; in this scenario, the failure plane is




governed by the Mohr-Column failure criteria of unsaturated soil mass, and is assumed to be parallel to the slope
surface (Fig.1). The expression of $F_s$ based on the shear strength formula of the unsaturated soil (Fredlund and
Rahardjo, 1993) and the infinite slope model can be expressed as follows:
$$Fs = \frac{\tan\varphi}{\tan\beta} + \frac{c + \psi\tan(\varphi^b)}{\gamma_t H_s \cos\beta\sin\beta} \tag{1}$$

Where $c$ is the cohesion force, $\varphi$ is the internal friction angle, $\varphi^b$ is related to the matrix suction (Which is close to
the internal friction angle $\varphi$ in the condition of the low matrix suction), $H_s$ is the soil depth, $\psi$ is the matrix suction
of the soil, which is a function of the soil water content described as follows (Van Genuchten, 1980):
$$S_e = \frac{\theta - \theta_r}{\theta_s - \theta_r} = \left[\frac{1}{1 + (\alpha \times \psi)^n}\right]^m \tag{2}$$

where $S_e$ is the saturation degree, $\theta_s$ is the saturated water content, $\theta_r$ is the residual water content, $\theta$ is the soil
water content of the current hour, $\alpha$, $n$ and $m$ are the parameters of soil-water characteristic curve, and $n=1-1/m$.

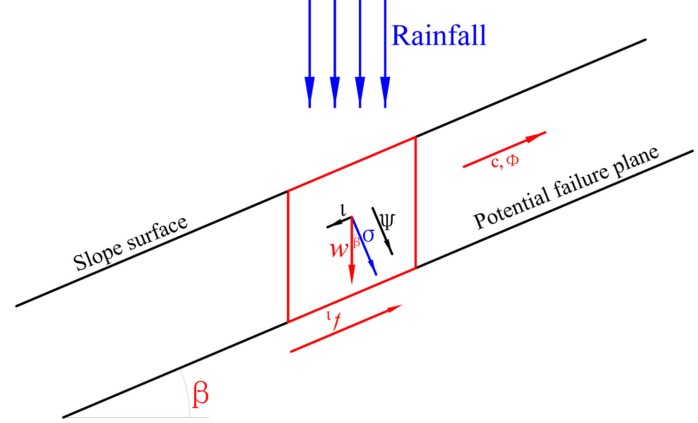


Fig.1 Infinite slope model for unsaturated soil in a slope
**2.2 Deterministic forecasting model using safety factor $F_s$**
The infinite slope model aims to calculate the safety factor $Fs$ to identify the stability of a slope. It has its basis
in a theoretical hypothesis (Apip et al., 2010), which can describe the mechanical process of shallow landslides
formation. This approach can give reliable results for each pixel as long as the soil mechanical parameters are
accurate. From a deterministic point of view, this physical framework can be briefly drawn as follows: for each
pixel in the forecast area, if $Fs \leq 1$ it's considered unstable, while pixels with $Fs > 1$ are considered to be stable.
Acquiring the values for the soil mechanical parameters necessary for the infinite slope model require the use
of field sampling or soil-texture based methods (Blondeau, 1973; Apip et al., 2010; Zhang et al., 2014a; Zhang et
al., 2014b). However, the precision of these methods are relatively low (Schmidt et al., 2008), thus subject to high
levels of uncertainty. Consequently, the seemingly deterministic infinite slope model based on soil mechanical
parameters of each pixel is in fact uncertain (Schmidt et al., 2008; Rossi et al., 2013). This will be reflected in the
safety factors $Fs$ of each pixel, leading to a situation in which, despite the advantages of the physical-based land-
slide forecasting model, it may be misleading if used in a deterministic way for real world applications.
This is not an issue for other landslide forecasting models. For example, although the input variables of the
contribution-factors-based forecasting model are also uncertain (Wei et al., 2007a) and thus it essentially belong to
statistical models (Zhang et al., 2014a) it successfully account for the relationship between uncertainties of input

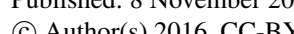

variables and results using fuzzy mathematics so that they are expressed as probabilistic forecasting for landslides.
The landslide probability is divided into five grades from 1[st] to 5[th] level, which represents a low, relative low,
medium, high and extremely high probability of occurrence of landslides, respectively. This forecasting result
conveys clearer landslide risk levels to the public (Wei et al., 2007b).
Due to the above reasons it is relevant to identify an effective relationship between the landslide probability
and uncertain input variables with uncertainty (cohesion force and internal friction angle) in a physics-based
probabilistic forecasting model.

### 2.3 Probabilistic forecasting model for shallow landslides

In order to link landslide probability to uncertain variables, the nature of this uncertainty should be quantita-
tively expressed in mathematical language. Then, a physical parameter associated with both, input variables and
landslide probability will be used to formalize the linkage.
The uncertainty of physical parameters can be described by a probability density function (Schmidt et al.,
2008). Accordingly, the uncertainties of cohesion force and internal friction angle are described here as uniform
probability distributions in the intervals of $c=U(c_{min}, c_{max})$, and $\varphi=U(\varphi_{min}, \varphi_{max})$, respectively. Then, Monte Carlo
method can be used to randomly extract cohesion force and internal friction angles from the two intervals $n$ times
in any forecasting step. This random approach is used to account for the uncertain nature of soil mechanical pa-
rameters. The detailed description of random extracting process is as follows: the extraction of the two parameters
is dependent on the a variable $r_i$ which is described as uniform probability distributions in the interval of $r_i=U(0,1)$,
the random values of cohesion force $c_i$ and internal friction angle $\varphi_i$ can be identified via the equations:

$$c_i = r_i(c_{max} - c_{min}) + c_{min} \qquad (3)$$

$$\varphi_i = r_i(\varphi_{max} - \varphi_{min}) + \varphi_{min} \qquad (4)$$

There, $c_{min}$ and $\varphi_{min}$ are lower borders of intervals of the two mechanical parameters expected values; $c_{max}$ and $\varphi_{max}$
are the upper borders. For any pixel in any forecasting step, a matrix $M_i$ can be generated after the $n$-times random
extraction process:

$$M_i = [c_i, \varphi_i] = \begin{bmatrix} c_1 & \varphi_1 \\ c_2 & \varphi_2 \\ c_3 & \varphi_3 \\ ... & .... \\ c_n & \varphi_n \end{bmatrix} \qquad (5)$$

Any element contained in $M_i$ has a specific physical meaning representing as a whole the physical phenome-
non of uncertainty.
Provided other parameters identified in Eq. 1, each set of $[c_i, \varphi_i]$ in $M_i$ can generate a safety factor
$Fs_i = [Fs_1, Fs_2, Fs_3, ..., Fs_n]$. The array of safety factors $Fs_i$ reflects $n$ possible stable states for a pixel under these
physical conditions. It's possible from there identify a failure probability by the number of $Fs_i \le 1$ (failure) in the $n$
different states in the form of a ratio $P$ ($P \in [0,1]$) of $Fs_i \le 1$ representing a tendency of a pixel to failure from sta-
bility.

$$P = \frac{Sum_{Fs<1}}{n} \qquad (6)$$

Larger $P$ values in Eq. 6 indicates a forecasting result favorable to a high occurrence probability of failure un-
der uncertain variables. This interpretation implies that a pixel will tend to one end failure when $P$ exceeds 50%
and its failure probability will only increase with larger values of $P$. Since $P$ is derived from series of random
(uncertain) variables $[c_i, \varphi_i]$ via Eq.1 and Eq. 6, and is also directly associates with the landslide probability, the
ratio ($P \in [0,1]$) of $Fs_i \le 1$ is a strong candidate for linking the landslide probability to the uncertain soil mechani-
cal parameters.



For the purposes of practical implementation of this forecasting model, *P* is divided into a series of reference
intervals in Table 1, the occurrence probability of shallow landslides increase from 1st interval to 5th interval of
*P*. Five grades of landslide warnings are defined accordingly and color-coded Table 1.
Table 1 Reference intervals for shallow landslides forecasting based in probabilistic safety factor

| Ratio intervals/% | $P < 20$ | $20 \leq P < 50$ | $50 \leq P < 60$ | $60 \leq P < 80$ | $80 \leq P < 100$ |
|---|---|---|---|---|---|
| Warning degree | 1 | 2 | 3 | 4 | 5 |
| Warning color | Colorless | Blue | Yellow | Orange | Red |

**3 Probabilistic shallow landslides forecasting method at regional scale**
**3.1 Gathering basic data necessary for landslide forecasting**
Topography is the main factor in shallow landslides. Nowadays, obtaining a DEM of precision adequate for
regional scale forecasting is straightforward. The DEM of the study zone is re-sampled into pixels with dimen-
sions according to the extension of the area. The parameters required to calculate the ratio *P* for each pixel from
the array of safety factors $Fs_i$ from a series of randomly extracted $[c_i, \varphi_i]$ are identified in Eq.1. In this case matrix
suction, which is associated with the soil water content, should be identified by hydrological process simulation.
The key data necessary for the hydrological process simulation include the spatial distribution of precipitation,
land use, soil type and NDVI. Precipitation data with the same solution of the DEM can be obtained by
re-sampling rainfall prediction from Doppler radar supplied by meteorological bureaus. Land use, soil type and
soil depth can be obtained from corresponding databases, all of which should be transformed into grid data with
the same solution of DEM. Other data necessary for stability calculations are slope angle for each pixel, parame-
ters from soil-water characteristic curve ($\alpha$, $m$, $n$), and soil mechanical parameters. Slope angles can be derived
from DEM using spatial analyst tools, parameters ($\alpha$, $m$, and $n$) of the soil-water characteristic curve are derived
from the different soil types within the pixel.
Regarding the identifications of soil mechanical parameters (cohesion force and internal friction angle), a rela-
tively reliable way such as field sampling or soil-texture based methods should be used to assign an initial basic
value to each pixel. Although these values include high uncertainty levels, they are used only as reference values
while setting intervals of $c=U(c_{min}, c_{max})$, and $\varphi=U(\varphi_{min}, \varphi_{max})$ (Raia et al., 2014). In this study, the lithology of the
study zone is derived from a geological map, and the mechanical parameters (cohesion force and internal friction
angle) of the corresponding lithology are identified using a rock mechanics handbook (Ye et al., 1991). Finally the
data is assigned to each pixel using the grid cells of the DEM as reference.
From Eq.3 and Eq.4, it is necessary to identify the lower and upper border of intervals of the soil mechanical
parameters. However, the exact values for lower ($c_{min}$ and $\varphi_{min}$) and upper ($c_{max}$ and $\varphi_{max}$) limits are very difficult
to determine. From currently published papers, there is no known theoretical or experimental method to solve this
issue. Raia et al. (2014) used variations of 1%, 10% and 100% around the values of cohesion force and internal
friction angle (from field tests) to get several intervals, showing that the forecasting effectiveness is significantly
improved by using a large variations. Consequently, this method applies a variation of 100% around the mean
value of these parameters for each pixel to set the corresponding lower and upper borders as follows:

$$c_{\text{random}} \in [0.5 \times c_{\text{origin}}, 2 \times c_{\text{origin}}] \tag{7}$$

$$\varphi_{\text{random}} \in [0.5 \times \varphi_{\text{origin}}, 2 \times \varphi_{\text{origin}}] \tag{8}$$

Where $c_{\text{random}}$ and $\varphi_{\text{random}}$ are the randomly extracted cohesion forces and internal friction angles, $c_{\text{origin}}$ and $\varphi_{\text{origin}}$
are the mean value of each pixel (in this case from the rock mechanics handbook (Ye et al., 1991)).
**3.2 Pixel level hydrological process simulation**
The simulation of hydrological processes in the soil, which includes rainfall interception, infiltration, and
evapotranspiration, is extremely complicate. However, in mountain conditions rainfall infiltration is the key factor



in the distribution of soil water content in soil underlying surface which simplify the analysis. In southwestern region of China slopes are almost unsaturated before the rainy season due to characteristic distribution of rainfall influenced by monsoon (Zhang et al., 2014b). The infiltration process in the vertical direction in unsaturated soil mass can be described by the 1D Richards's equation (1931):

$$\frac{\partial \theta}{\partial t} = \frac{\partial}{\partial z}[D(\theta)\frac{\partial \theta}{\partial z}] - \frac{\partial K(\theta)}{\partial \theta} \tag{9}$$

Where $\theta$ is soil water content, $D(\theta)=K(\theta)/(d\theta/d\psi)$ is the hydraulic diffusivity, $\psi$ is the suction of unsaturated soil, $z$ represents the soil depth, which is positive along the soil depth and have the topsoil as the origin point, $K(\theta)$ is the hydraulic conductivity. The matrix suction is the dominant external force to drive the water movement in unsaturated soil mass, which can be calculated from Eq. 2.

Infiltration upper border: If the topsoil is unsaturated, it has a strong infiltration capacity (Lei et al., 1988). Then, while the rainfall intensity is less than the infiltration capacity of the topsoil, all precipitation will infiltrate into topsoil without any runoff. In this scenario, the infiltration border is governed by Eq. (10):

$$-D(\theta)\frac{\partial \theta}{\partial z} + K(\theta) = R(t), \quad t > 0, z = 0 \tag{10}$$

Where $R(t)$ is the rainfall intensity at time $t$. Here, the part of precipitation that exceeds the capacity of infiltration of the topsoil will transform into runoff (no water storage above topsoil). In this case the topsoil of a pixel is considered saturated. Thus, the Eq.10 that governs infiltration upper border is transformed into the equation of $\theta=\theta_s$ (Lei et al., 1988). There $\theta_s$ is the saturated moisture corresponding to the soil type.

Infiltration bottom border: It has been experimentally demonstrated that the soil water content beyond a soil depth of 40 cm is barely influenced by rainfall infiltration (Cui et al., 2003). Consequently a region with a groundwater level near the surface of the soil has hydrological characteristics in which rainfall infiltration can hardly induce any groundwater level variation. In this case, it is reasonable to ignore the water exchange process between the lower boundary and groundwater (Zhang et al., 2015).

An implicit finite difference method is used for discretization of the 1D differential equation of water movement. The calculation time $t$ is segmented into several intervals with the same time gap $\Delta t$, and the soil depth $L$ of each pixel is segmented into soil layers with the same depth $\Delta z$.

Identifying the initial soil water content is an important issue during the hydrological simulation process. However, this value cannot be directly determined at any given time for a large region due to complex rainfall infiltration and evapotranspiration interactions. In the case of southwestern China, the winter is generally a relatively dry season thus; the soil water content value of the topsoil is very low, close to the residual water content of the soil type (Zhang et al., 2014b). This situation is exploited setting the simulation time to start on January 1 of the forecasting year (driest month in winter), which allows the use of the residual water content corresponding to the soil type as and the initial value of the topsoil water content. Measured meteorological data from January 1 are then feed to the simulation, which allows for a relatively accurate initial value of soil water content for the landslide forecasting. Each simulation step take also into account the rainfall interception and evapotranspiration processes by means of the algorithm of distributed hydrological model GBHM (Yang et al., 2002).

After the hydrological simulation process identify the initial soil water content of each pixel, the simulation focuses on the extraction of key hydrological parameters (soil water content and matrix suction) necessary for the stability calculation of each pixel using the expected rainfall from Doppler radar forecasting. During this last stage in the simulation in which landslide forecasting is performed, the evapotranspiration processes is not considered since this period is typically short, with rainfalls, negligible sunshine and lower temperatures.

### 3.3 Probabilistic landslide forecasting at pixel level

During the forecasting stage, the hydrological parameters (soil water content and matrix suction) of each pixel





in each forecasting step $\Delta t$ are extracted via hydrological process simulation. Then the ratio $P$ is computed for
each pixel in several steps as follows: (1) The Monte Carlo method is used to extract the cohesion force and the
internal friction angle $n$ times from the corresponding intervals ($c=U(c_{min}, c_{max})$, and $\varphi=U(\varphi_{min}, \varphi_{max})$) of each pixel;
(2) The safety factor $F_s$ of the pixel is calculated after each extraction, using the soil mechanical parameters and
the hydrological parameters only related to time as inputs of Eq.2; (3) Once the Monte Carlo process end, the total
times $Sum_{Fs<1}$ of $F_s<1$ is obtained, and the ratio $P$ of $F_s<1$ is calculated by Eq.6; (4) Finally the interval of Table 1
where ratio $P$ is located according to its value is assigned to the pixel as the early warning information to be
broadcasted.
After completing this process for all pixels within the forecasting region, the whole calculation at time $t$ is fin-
ished, meanwhile a map of landslide warning degrees in the forecasting region will be generated at the end of each
forecasting step. Such maps can then be used by the forecasting bureau of the region to issue landslide warnings to
hazard mitigation units and the public.
**4 Verification of the probabilistic landslide forecasting model**
**4.1 Study zone**
The Wenchuan earthquake region within Sichuan province, China is chosen as the study zone in this study
(Fig.2). In this region, at 14:28 PM (Beijing time) on May 12$^{rd}$ 2008, an Ms 8.0 earthquake occurred. Massive
potential unstable slopes were left after this earthquake, which are known to readily evolve into shallow landslides
by rainfall infiltration (Zhang et al., in Pres.). The close relationship between rainfall and landslides in this region
has been demonstrated by the short lag time of landslides and its strong correlation to rainfall time (Tang, 2010).
The same study established that landslide events within the earthquake region are mainly in the form of shallow
landslides (Tang, 2010). Tang (2010) also pointed out that shallow landslides will be active within Wenchuan
earthquake region at least for the next ten years. Such conditions make this region ideal for implementation of
shallow landslides forecasting models.

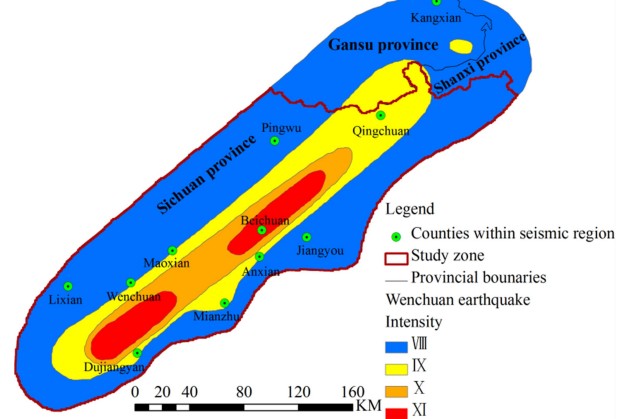


Fig.2 Study zone and intensity distribution of Wenchuan earthquake
**4.2 Rainfall process and related landslide events used for testing**
The chain of events in the Wenchuan earthquake area that ended in disastrous landslides in July 9$^{th}$ of 2013
was chosen to evaluate the proposed landslide probabilistic forecasting method. These events started with heavy
rainstorms in the area during the days from July 1$^{th}$ to July 8$^{th}$ of 2013. As the rainfall measured by the weather
stations within the area shows (Fig.3), the maximum accumulated precipitation during these days reached 317.7
mm, which become a key contributing factor for the landslide events of July 9$^{th}$ of 2013.

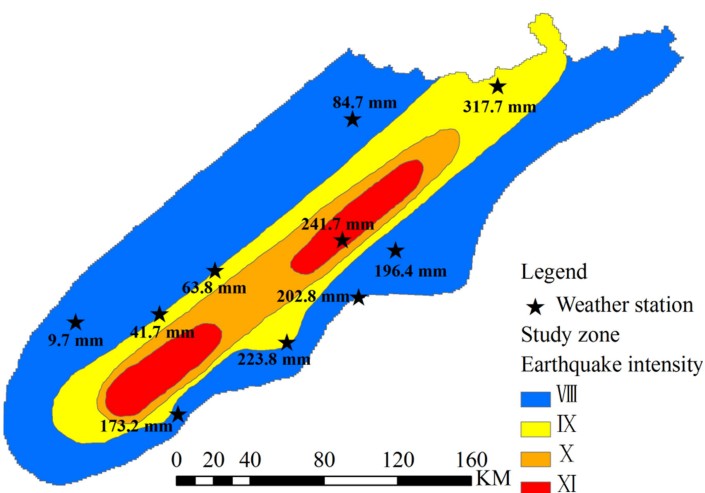


Fig.3 Total rainfall from 1$^{st}$ to 7$^{th}$ of July 2013

On July 9$^{th}$ of 2013, there was no evidence of decreasing rainfall intensity, on the contrary all evidence sug-
gested heavier rainfalls. Records from the rainfall forecasted by Doppler radar provided by the weather bureau of
Sichuan province on that day, predicted a maximum 24-hour total precipitation within the earthquake region of up
to 498 mm (Fig.4). Accordingly, the Weather Bureau of Sichuan province published red color warning signals
(which are the highest alert degree) for some locations within the study region. On that day, 176 landslide events
were reported within the study region (Fig.4) leading to casualties and serious economic losses for local residents
(Zhang et al., 2014b). This typical landslide disaster triggered by intense rainfall is ideal to evaluate the main as-
pects of the implementation of the proposed probabilistic landslide forecast model at regional scales.

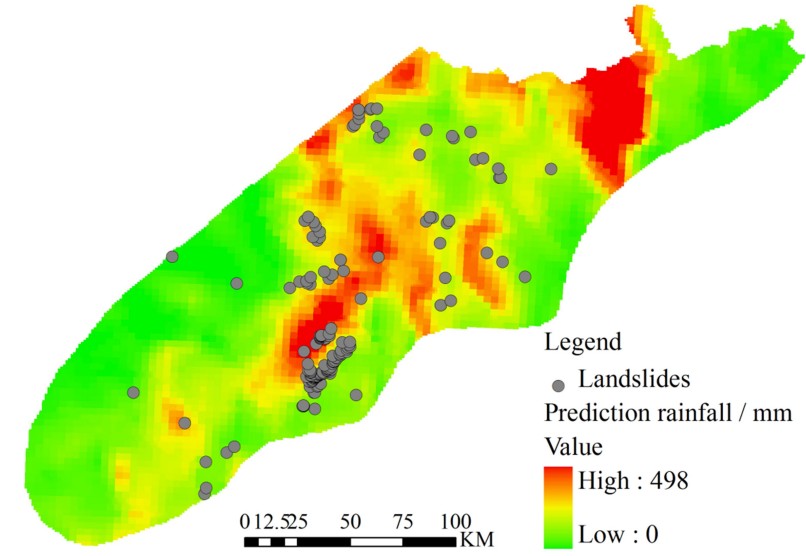


Fig.4 Distribution of rainfall-induced landslides within Wenchuan earthquake region on July 9$^{th}$ of 2013
**4.3 Gathering of basic data of study zone**
The topography of the study region (Fig.5) was described by 125 m $\times$ 125 m DEM. This way, the study


region with an area $3.14 \times 10^4$ km$^2$ was segmented into 6965505 pixels. A data matrix with 2576 rows and 2704
columns was created from the DEM and saved in text format. The basic data for hydrological process simulation
and stability was resampled to correspond to the same resolution of the DEM and saved as text matrices with the
same dimensions.

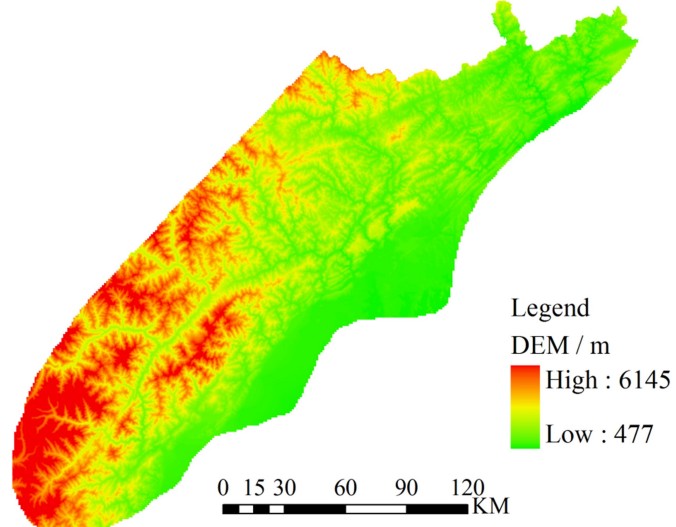


Fig.5 DEM of Wenchuan earthquake area
**4.3.1 Data for hydrological process simulation**
The process of rainfall interception due to vegetation influence within the study region is taken into account
using NDVI values. Generally, the vegetation, and thus the values of NDVI vary with the variation of land uses
and seasons. In this case, NDVI values from the same reason of the adjacent year are considered reasonably close,
since the distribution of land uses within a region is relatively stable. The monthly NDVI distribution over the
study region in the precedent year (2012) was used to adjust for canopy rainfall interception during the hydrolog-
ical process simulation (Fig.6).

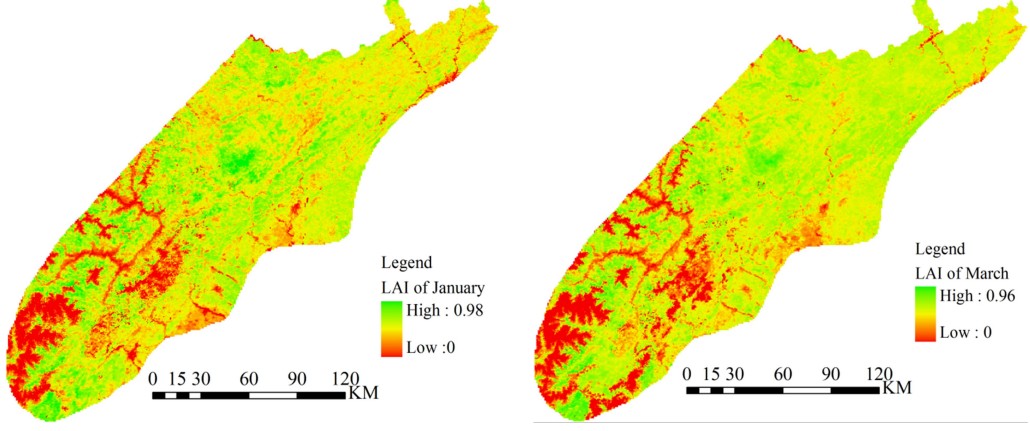




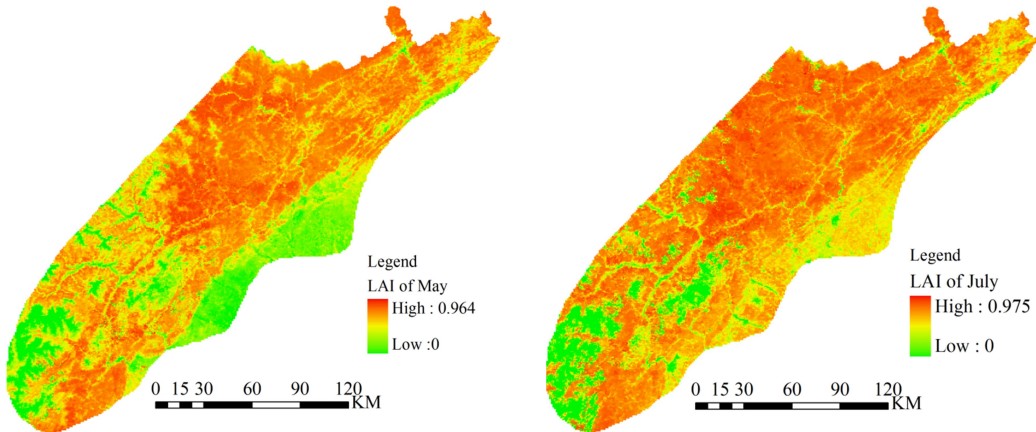


Fig.6 Distributions of the LAI within the study zone
Other data required, such as land use (Figure 7 (a)), soil type (Figure 7 (b)), and the soil depth for Wenchuan
earthquake region was obtained from the FAO database (http://www.fao.org/geonetwork/srv/en/main.home). The-
se data was processed using GIS functions so that they correspond to the pixels of the DEM.

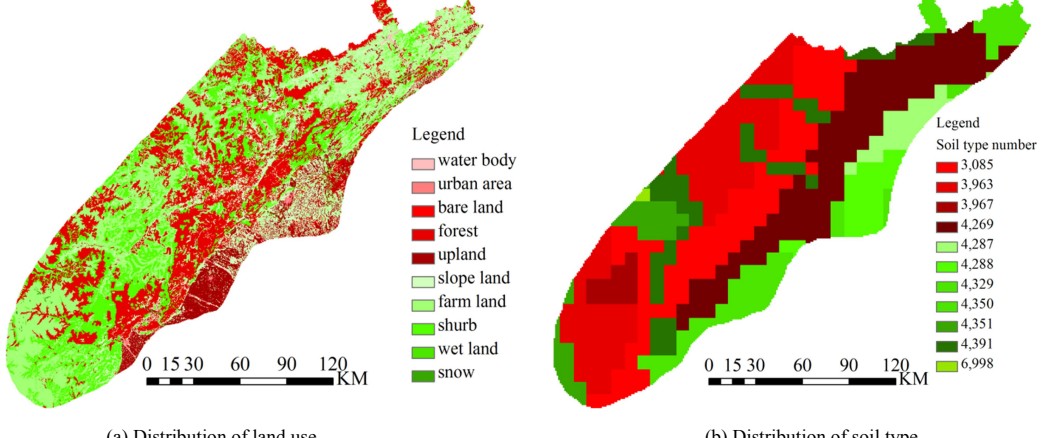


(a) Distribution of land use             (b) Distribution of soil type

Fig. 7 Information of land uses and soil types within the study zone
The physical parameters of the soil required for the simulation of rainfall infiltration in the vertical direction
were determined by the land use and standard soil types within the study region. The soil thickness ranged from 1
to 4 m, soil depths of 1 m accounts for 44.1% of the study area, while deeper soils cover the remaining 55.9%.
Each pixel was divided into 10 layers (along the soil depth in the vertical direction) during the discretization pro-
cess. There are 10 soil types in the area (shown in Fig. 7b). Their relevant physical properties are listed in Table 2.
Table2 Soil-water parameters for hydrological simulation

| Soil type code | Saturated moisture | Residual moisture | Parameters of curve | | Saturated hydraulic conductivity(mm/h) |
| --- | --- | --- | --- | --- | --- |
| | | | *Alpha* | *n* | |
| 3085 | 0.48278 | 0.07768 | 0.01896 | 1.40474 | 22.78608 |
| 3963 | 0.47303 | 0.07347 | 0.01796 | 1.42367 | 22.46508 |
| 3967 | 0.52726 | 0.08259 | 0.01867 | 1.41453 | 35.97075 |
| 4269 | 0.45649 | 0.06905 | 0.02306 | 1.55872 | 32.68625 |
| 4287 | 0.44596 | 0.07343 | 0.01971 | 1.47235 | 19.30871 |
| 4288 | 0.43797 | 0.07175 | 0.02064 | 1.53067 | 24.80996 |
| 4329 | 0.45049 | 0.07957 | 0.01604 | 1.44517 | 9.307170 |
| 4350 | 0.47990 | 0.07435 | 0.02156 | 1.42176 | 22.51646 |



| 4351 | 0.48278 | 0.07723 | 0.02040 | 1.41974 | 21.61279 |
| 4391 | 0.42784 | 0.06439 | 0.01623 | 1.63524 | 23.91267 |
| 6998 | 0.46154 | 0.06817 | 0.01770 | 1.46884 | 23.60925 |

**4.3.2 Data for calculation of slope stability**

The Eq.1 indicates that matrix suction, cohesion force, and internal friction angle are the key mechanical parameters influencing the slope stability. Simulation of the hydrological process is used to obtain the matrix suction of soil mass as a function of the soil water content as shown in Eq. 2. Cohesion forces and internal friction angles for each pixel are determined according to lithology map and the rock mechanical handbook (Fig.8); these two maps were updated from the old database (Liu et al., 2016). These mechanical values are then used as a basic reference for constructing intervals of these parameters ($c=U(c_{min}, c_{max})$, and $\varphi=U(\varphi_{min}, \varphi_{max})$) for each pixel.

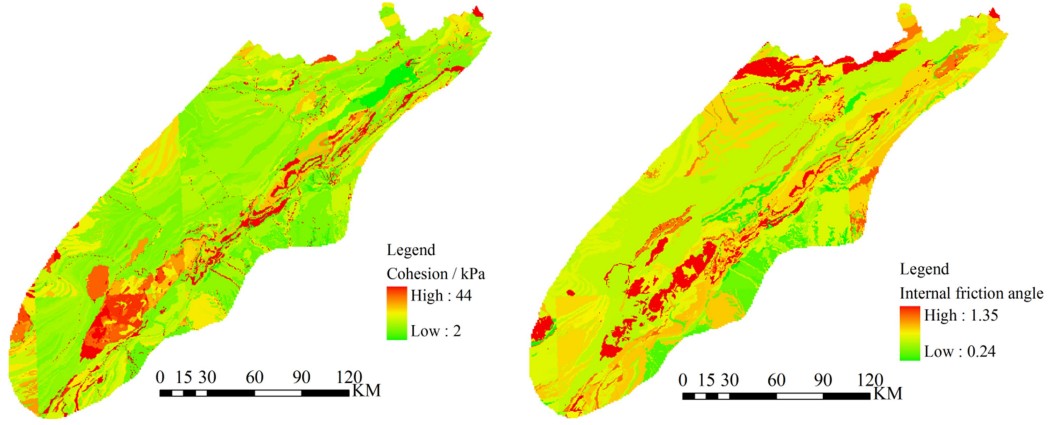

(a) Distribution of cohesion forces                    (b) Distribution of internal friction angles

Fig.8 Mechanical paramters of soil used for calculation of slope stability

**4.4 Forecasting results**

The landslide probability in Wenchuan earthquake region on July 9, 2013 was calculated, along with color-coded warnings for each pixel according to Table 1. This forecast covered 24 time nodes (hourly forecasts) covering the whole day. Two representative time nodes (at 6:00 AM and 15:00 PM) are chosen from the 24 h forecasting results for further analysis (figure 9). The detailed forecasting results are listed in Table 3. These details denote low variation in the forecast for these time nodes.

Table 3 Quantity of pixels with warning information

| ` | | Blue | Yellow | Orange | Red |
|---|---|---|---|---|---|
| pixel | 6:00 AM | 534 | 150 | 332 | 699 |
| count | 15:00 PM | 527 | 158 | 321 | 704 |

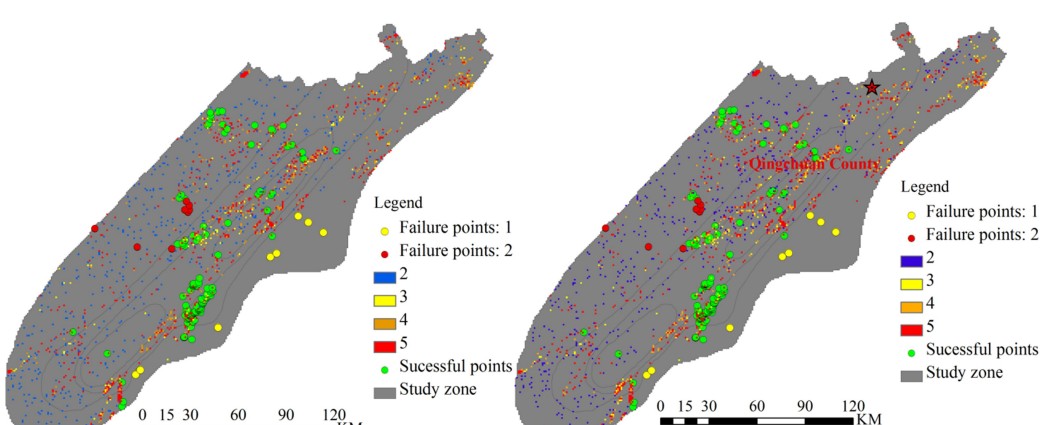


(a) Forecasting information at 6:00                     (b) Forecasting information at 15:00

Fig.9 Landslide warning maps for Wenchuan earthquake region at two representative time nodes.
Colored points in fig. 9 represent landslide disasters occurred on July 9, 2013. Green points represent land-
slides located in pixels forecasted with high degree of probability of landslides (orange-red), thus they are consid-
ered successfully forecasted or true positives (159 events). The other 17 events represented by yellow and red
points denote landslide events in low warning areas, which are considered as failed-forecasted landslides or false
negatives. These numbers indicate a missing-prediction rate of the new proposed forecasting model of about

336 9.7%.

Further analysis of these failures indicated that in some cases, the maximum slope angle of the corresponding
pixel reported by the DEM is less than 4 degrees (yellow points). Furthermore, 4 of these pixels have slope angles
equal to 0 from the DEM. These small angles are for practical effect equal to flat terrain. In this scenario the
probabilistic forecast model is unable to predict any unstable state, even during a more serious rainstorm. Howev-
er, the real occurrence of landslide events at these locations indicates further analysis is necessary. In this case, the
most probable cause of this situation is the generalization process associated with the resolution of the DEM. It is
well known that increasing the size of the pixel tends to lower the estimated slope value, which in turn will raise
the failure prediction rate of models with high dependence on accurate slope values. A straightforward solution to
this problem is to further reduce the size of the pixel, which will in turn represent the real slope angle more accu-
rately. This solution however will drastically increase the computing time. As reference, the current matrix dimen-
sions of 2576×2704 (for 125 m pixel size) represent the limit for a regular workstation when the data is not parti-
tioned.
There is still 8 prediction failures (marked by red dots) unexplained. These are considered to be related to oth-
er aspects of the probabilistic forecasting model and unaccounted uncertainties. Detailed forecasting information
about the landslide events in this study is listed in Table 4.
Table 4 Detailed forecasting analysis

| landslides | Successful predicted landslides | Failure to predict land-slides due to DEM imprecision | Failure to predict land-slides due to model imprecision | Failure rate |
|---|---|---|---|---|
| 176 | 159 | 9 | 8 | 9.7% |

The false prediction (false positives) rate for the probabilistic forecast model is high. The Fig. 9 shows high
warning degrees concentrated around Guangyuan City and Qingchuan County, where landslide events did not
occur. Looking at Fig. 3, the accumulative precipitation within Guangyuan City during the days of July 1st and 7th
are 317.7 mm according to the local weather station. This implies initial soil water contents in the region close to

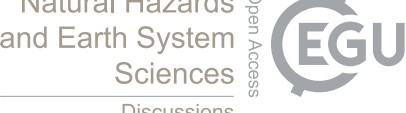



saturation levels just before the forecasting time. Additionally, the cumulative precipitation predicted from the
Doppler radar reached more than 470 mm in Guanyuan City. Under the action of such a combination of strong
antecedent rainfall and forecasted rainfall, it is reasonable to expect high concentration of landslides (forecasted
by the probabilistic model with different warning colors). Although the measured rainfall data for July 9th was not
available for this study, indirect information (absence of report of landslides and other phenomena associated with
heavy rainfall, even with notable initial soil water content levels) indicates the real precipitation on July 9th was
much smaller than forecasted from Doppler radar. Adding the known tendency of Doppler radar forecasts to over-
estimate rainfall, it is reasonable to consider the precision of Doppler radar rainfall as a key factor influencing the
high false prediction rates of the proposed probabilistic forecasting model.
**5 Discussions**
The general rule for the evolution of a slope from stability to failure is that the failure probability should in-
crease as the rainfall process continues since increasing soil water content will decrease the suction matrix. This
rule implies a forecasting result at 15:00 PM with more unstable pixels than the result at 6:00 AM. However, both
of them are relatively close.
The distribution map of initial soil water content at 24:00 on July 8th, shown in Fig. 10, indicates significant
effects of accumulated rainfall for landslide forecasting, the topsoil of some areas are even in saturated conditions
(this means that only the topsoil was saturated rather than the whole soil layer). The total saturated pixels within
study region are 532.

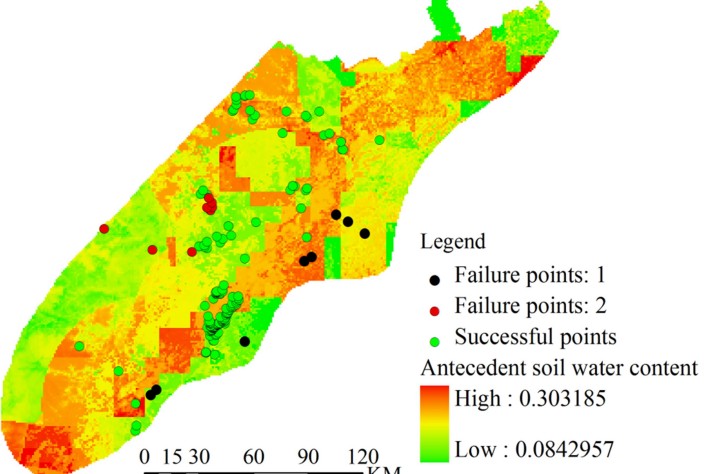

Fig.10 Intial conditions for landslide forecasting
Under these initial conditions, the mechanism of the runoff-infiltration process indicates that significant
amount of precipitations will transform directly into runoff as the soil water content value of topsoil increases. In
this case study, these high levels of initial soil water content attributed to strong antecedent rainfalls leads to lower
variation rate of soil water content at pixel level. In this scenario, the variation of soil water content tends be gen-
tle even during long and intensive rain, while excess water contribute mainly to the runoff process. This chain of
events may explain the lack of clear evolution in the forecast in this particular study.
To further confirm this analysis, a new hydrological simulation was run in which the antecedent precipitation
is ignored. The initial soil water content of each pixel for landslide forecasting was directly assigned with the
residual soil water value according to the corresponding soil type (assuming a completely dry soil). All other pa-
rameters, including predicted rainfall from Doppler radar remained unchanged from the previous simulation. The




forecast results at 6:00 AM and 15:00 PM under these new conditions are shown in Fig. 11 and Table 5. It is easy
to observe differences between forecasting times, with quantity of unstable pixels at 15:00 PM larger than at 6:00
AM as expected. In this case, the low level of initial soil water content allows for strong infiltration process in the
topsoil, which in turn leads to high variation rates for soil water content in each pixel, reflected in the differences
of forecasting aligned with the expected evolution of the slope failure process.

Above analysis not only explain why there is not big difference between 6:00 AM and 15:00 PM forecasts dur-
ing a high intensive rainstorm. It also to stress the relevance of the initial soil water content (or the effective ante-
cedent rainfall) for any physically based landslide forecast model. A reliable method to calculate the initial soil
water content can significantly influence the results of landslide forecasting models.
Table 3 Quantity of pixels with warning information, without considering the influence of antecedent soil water content

| Warning colors | | Blue | Yellow | Orange | Red |
|---|---|---|---|---|---|
| pixel | 6:00 AM | 229 | 106 | 237 | 325 |
| count | 15:00 PM | 328 | 128 | 290 | 586 |

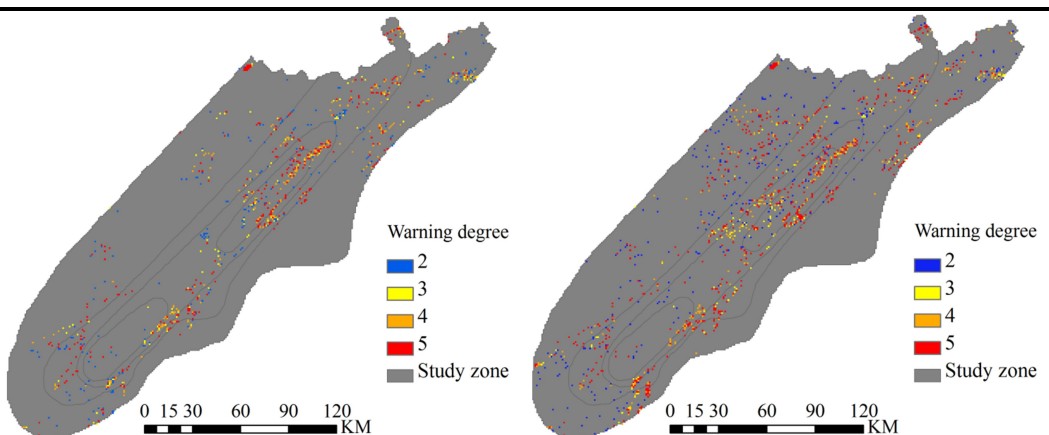


Fig.12 Forecasting results without considering the influence of the antecedent soil water content

Another issue is that most published physical models for landslide forecast such as the SLIP and TRIGRS
models (Montrasio et al., 2011; Tsai and Chiang, 2012) overestimated the probability of landslide occurrence at
regional scales. This proposed physics-based probabilistic forecasting model is also affected by this problem.
From the point of view of input parameters, three key factors can lead to this high false prediction rate. (1) The
soil mechanical parameters can only be obtained indirectly at regional scales, which greatly increase uncertainty.
Consequently, it is impossible to guarantee the correspondence of the fixed mechanical values at pixel level with
the actual values in nature, even using large intervals of soil mechanical parameters such as in this paper. Under-
estimating these values increase the probability to identify the corresponding pixel as unstable, which contribute
to high false prediction rates. (2) The nature of DEM models implies that a pixel identified as unstable by a pixel
based forecasting model may not really represent an unstable slope in nature. A slope may contain several pixels
of which only a few are unstable, or more likely at regional scales, a pixel may include several slopes. In this sce-
nario isolated unstable pixels can contribute to high false prediction rates. (3) The precision of short term rainfall
forecasting is the last factor that can contribute to high false prediction rates. This is relevant in this study in which
rainfall forecasts from Doppler radar overestimated the expected rainfall in some areas.
**6 Conclusions**

The extreme complexity of the landslide formation process conditions that even physics-based forecasting
models are unable to model the slope instability with 100% of confidence. However, the uncertainty of some input




variables (e.g., soil mechanical parameters) is responsible for a significant part of this situation. This research adopted a probabilistic approach to express this uncertainty using Monte Carlo simulation. A single parameter (the ratio $P$) was devised to couple the uncertain nature of input variables with shallow landslides forecasting. Furthermore, a regional physics-based probabilistic shallow landslide forecasting model was developed around this parameter. The proposed model does not eliminate uncertainty; it manages it by explicitly introducing it into the model expressing the forecast directly in probabilistic form. Our tests shown that this approach increases the forecast precision (true positives) in real conditions, which is cardinal to protecting the public from catastrophic consequences of shallow landslides and other associated disasters (such as debris flows).

It must be noted that the complexity of landslide forecasting is not limited to the uncertainty of physical soil properties, this research points to the initial soil water content as another key variable extremely difficult to identify accurately at regional scales. The model proposed in this paper implements a simulation of the hydrological processes occurring in the soil to estimate this value. Such simulation is time intensive, which is unfavorable for real world applications. Future research should focus in efficient methods for identification of soil water content at regional scales, which is a difficult but worthy challenge.

The goal of developing this physics-based probabilistic forecasting model is to serve for regional landslide disaster mitigation. Detailed resolution data, which in case of DEMs is readily available, are not always straightforward solutions for better forecasting results at this scale. In this case higher DEM resolution will improve the efficiency of the model failure prediction rates at individual pixel level due to better slope representation. However, it will also increase the time and resources required by the model to produce usable results. A balance point between pixel-level precision and operational efficiency is required for the proposed model in order to make it more suitable for regional operation.

**Acknowledgement:** This work was supported by Science and Technology Service Network Initiative (No: KFJ-SW-STS-180), the Science and Technology Support Project of Sichuan Province (No. 2015SZ0214), the Risk assessment on geohazards induced by extreme rainfall (CCSF201428), and hydrometeorological forecasting project from National Meteorological Center of China Meteorological Administration.

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
