# Peer review of "A physics-based probabilistic forecasting model for rainfall-induced shal low landslides at regional scale"

_Natural Hazards and Earth System Sciences, 2016_

## Short Comment (SC1) · 28 Nov 2016

Concerning 1D Richards's equation (RE), authors could cite in their work:

De Luca, D.L., Cepeda, J.M. Procedure to obtain analytical solutions of one-dimensional richards' equation for infiltration in two-layered soils. (2016) Journal of Hydrologic Engineering, 21 (7): 04016018 DOI: 10.1061/(ASCE)HE.1943-5584.0001356

In this paper the goal was to provide a procedure aimed at obtaining analytical solutions for one-dimensional (1D) RE in two layer soils, particularly appropriate for DEM-based models.

[Figure]

2016.

---

## Referee Comment (RC1) · Anonymous Referee #1 · 23 Feb 2017

Zhang et al present a large scale analysis of landslide hazards by connecting hydrological process simulation and a probabilistic slope stability model. The innovative content of the work lies in the application of a Monte Carlo approach to assess the uncertainty of geotechnical model parameterization (cohesion and internal friction angle). The objective of the study is very ambiguous as landslide probability calculation is carried out for a large area (31000 km$^2$) on a daily basis. The spatial resolution is 125 m. Consequently several simplifications are necessary. The success rate of landslide prediction for a rainstorm event in 2013 is high, as well as the false prediction rate. In my opinion the manuscript is well written and has an adequate structure. The authors do a good job in explaining what they did and the probabilistic approach is of interest for the

research community. For this reason I recommend acceptance with minor revisions.

The following specific comments cover mainly English style and grammar and may not be exclusive: L 88: Mohr-Coulomb is misspelled L 92: c is a stress L 92: "(which)" instead of "(Which)" L 133: "dependent on the variable r" instead of "dependent on the a variable r" L 145: add "to" before "identify" L 193-194: I suggest to re-formulate this sentence. L 228: "takes" instead of "take" L 305: the depth of the shear plane has a crucial influence on the FOS. It seems that the depth of the shear plane was assumed to equal the depth of the soil. Is this correct? Please comment (also in section 5) on the consequences of this assumption (sensitivity of model outcome) and on the accuracy of the spatial soil depth distribution assumed in this study. L 307: please explain the discretization process in more detail Section 4: add information of the size of the investigation area could be given earlier. Figure 9: right: what does the star mean?

---

## Referee Comment (RC2) · Anonymous Referee #2 · 15 Mar 2017

This manuscript proposed a physical-based probabilistic forecasting model for rainfall-induced shallow landslides at regional scale, in which only cohesion and tangent of frictional friction were considered as uniform probability distribution model for per pixel, and other parameters are fixed. Because the cohesion and friction angle always are not uniform distributed within two limits, the proposed method is not based physically as called by the authors. In addition, please consider the following comments if the authors revised the manuscript. 1. As expressed in Eq. 1, the safety factor is influenced by phi_b related to the matric suction. How this parameter is determined in the calculation? 2. How about the initial water content along the depth? 3. The distribution of cohesion and internal friction angle in Fig. 8 should be consistent with the soil type

in Fig.7, but it is not so now.

---

## Author Comment (AC1) · 16 Mar 2017

Dear Davide Luciano: Thanks a lot for your kind suggestion to our paper, and I think that your paper "Procedure to obtain analytical solutions of one dimensional Richards' equation for infiltration in two-layered soils" is very impressive and interesting after carefully reading. We think this paper is outstanding and suitable for our paper reference, and we will cite your paper if the authors get the revision chance. Inspired by your paper, we will try to use your analytical solutions of one-dimensional Richards' equation for comparing our numerical solution in the next following studies. Thanks again for your paper.

[Figure]

Please also note the supplement to this comment:
http://www.nat-hazards-earth-syst-sci-discuss.net/nhess-2016-348/nhess-2016-348-AC1-supplement.pdf

---

## Author Comment (AC2) · 16 Mar 2017

Dear Reviewer,

Thanks a lot for your kind comments on our paper, I will reply your comments one by one, as follows:

(1) Reviewer: L 88: Mohr-Coulomb is misspelled.

Authors: Thanks for your careful checking, the Mohr-Coulomb was misspelled. We will use "Mohr-Coulomb" to instead of "Mohr-Column" in our paper.

(2) Reviewer: L 92: c is a stress.

Authors: Yes, c is a stress. We will use "c is a stress" to instead of "c is the cohesion force".

(3) Reviewer: L 92: "(which)" instead of "(Which)"

Authors: we will use "which is close to the internal friction angle $\varphi$" to instead "Which is close to the internal friction angle $\varphi$".

(4) Reviewer: L 133: "dependent on the variable r" instead of "dependent on the a variable r"

Authors: Thanks for your careful checking, the authors made a mistake, we will delete the word "a", and use the finial sentence: "dependent on the variable r".

(5) Reviewer: L 145: add "to" before "identify"

Authors: Thanks for your careful checking, we missed this word "to" and we will add this word before "identify".

(6) Reviewer: L 193-194: I suggest to re-formulate this sentence.

Authors: Yes, these two sentences are indeed not clear to understand. The authors will rewrite these sentences in the next following revision process.

(7) Reviewer: L 228: "takes" instead of "take"

Authors: The authors will use "takes" to instead of "take".

(8) Reviewer: L 305: the depth of the shear plane has a crucial influence on the FOS. It seems that the depth of the shear plane was assumed to equal the depth of the soil. Is this correct? Please comment (also in section 5) on the consequences of this assumption (sensitivity of model outcome) and on the accuracy of the spatial soil depth distribution assumed in this study.

Authors: if the depth of the shear plane was assumed to equal the depth of the soil, it is indeed not correct. In our model, we actually calculated FOS of each layer within

each pixel. As described in L 307, each pixel was divided into 10 layers with the same soil depth. For example the depth of each layer was equal to 0.2m if the total soil depth was 2m. So if the FOS of ith layer within a pixel was less than 1, then our model will consider that the instable depth was i*0.2m. The authors are so sorry for not clearly describing this issue and made the reviewer misunderstanding; we will add some detailed information to clear it.

(9) Reviewer: L 307: please explain the discretization process in more detail

Authors: According to the finite difference principle, the larger number of divided soil layers within one pixel, the more accurate simulating results for water movement between each soil layer. So the water soil content that has an important influence on the FOS will be more accurate, which will finally influence the accuracy of the model outcome. However, confined by the computer capacity, the soil layer of each pixel was set to be 10 in our paper, which is a general practice by using a certain layer number to discretize the soil depth. We will add some more details in L307 to explain the discretization process.

(10) Reviewer: Section 4: add information of the size of the investigation area could be given earlier.

Authors: we will add the size of the investigating area in Section 4.1.

(11) Reviewer: Figure 9: right: what does the star mean?

Authors: The authors are sorry for this mistake; we will delete this red star in Fig.9b.

Please also note the supplement to this comment:
http://www.nat-hazards-earth-syst-sci-discuss.net/nhess-2016-348/nhess-2016-348-AC2-supplement.pdf

---

## Author Comment (AC3) · 16 Mar 2017

Dear Reviewer,

Thanks a lot for your kind comments on our paper, I will reply your comments one by one, as follows:

(1) Reviewer: As expressed in Eq. 1, the safety factor is influenced by phi_b related to the matric suction. How this parameter is determined in the calculation?

Authors: $\Phi b$ is the parameter relating to the matrix suction, which is close to the internal friction angle $\varphi$ in the condition of the low matrix suction. And low matrix suction means high soil water content; this situation is favorable to landslide. So in our paper, $\Phi b$ is

set to be equal to the internal friction angle $\varphi$.

(2) Reviewer: How about the initial water content along the depth?

Authors: Initial soil water content was based on the residual water content of each soil type. If the residual water content of a soil type is equal to w, then any pixel belonging to this soil type will be assigned this value. This means that the initial water content of each layer was assumed to be w, namely uniform distribution along the depth. This distribution has some drawbacks, for example, the deeper soil layer may have higher soil water content, but we cannot identify and have to use the above easy identifying method.

(3) Reviewer: The distribution of cohesion and internal friction angle in Fig. 8 should be consistent with the soil type in Fig.7, but it is not so now.

Authors: If cohesion and internal friction angle was derived from the soil type, the distributions should be same. However, cohesion and internal friction angle were determined based on the lithology map and the rock mechanical handbook, this is the reason why their distributions are different.

Please also note the supplement to this comment:
http://www.nat-hazards-earth-syst-sci-discuss.net/nhess-2016-348/nhess-2016-348-AC3-supplement.pdf
* * *

---

## Author Response (AR1)

Dear Editor and Reviewers,

Thanks a lot for your kind comments on our paper. Now we have amended this manuscript according to the advice and used the track changes mode in MS to highlight modifications in this manuscript. Any change was also marked by yellow color.

We gave the detailed explanations about the revisions as follows:

**Reviewer 1:**

**(1) Reviewer: L 88: Mohr-Coulomb is misspelled.**

**Authors:** We have used "Mohr-Coulomb" to instead of "Mohr-Column" in our paper, location is L88.

**(2) Reviewer: L 92: c is a stress.**

**Authors:** We modified this sentence as follows: "a stress and can be named of the cohesion force". Location is L92.

**(3) Reviewer: L 92: "(which)" instead of "(Which)"**

**Authors:** we use "which" to instead "Which". Location is L93.

**(4) Reviewer: L 133: "dependent on the variable r" instead of "dependent on the a variable r"**

**Authors:** we deleted the word "a", and use the finial sentence: "dependent on the variable r". Location is L137.

**(5) Reviewer: L 145: add "to" before "identify"**

**Authors:** We added this word before "identify". Location is 151.

**(6) Reviewer: L 193-194: I suggest to re-formulate this sentence.**

**Authors:** The authors have rewritten these sentences in L199 and L200.

**(7) Reviewer: L 228: "takes" instead of "take"**

**Authors:** The authors used "takes" to instead of "take". Location is L234.

**(8) Reviewer: L 305: the depth of the shear plane has a crucial influence on the FOS. It seems that the depth of the shear plane was assumed to equal the depth of the soil. Is this correct? Please comment (also in section 5) on the consequences of this assumption (sensitivity of model outcome) and on the accuracy of the spatial soil depth distribution assumed in this study.**

**Authors:** if the depth of the shear plane was assumed to equal the depth of the soil, it is indeed not correct. In our model, we actually calculated FOS of each layer within each pixel. As described in L 307, each pixel was divided into 10 layers with the same soil depth. For example the depth of each layer was equal to 0.2m if the total soil depth was 2m. So if the FOS of $i^{th}$ layer within a pixel was less than 1, then our model will consider that the instable depth was i*0.2m. The authors are so sorry for not clearly describing this issue and made the reviewer misunderstanding; we will add some detailed information to clear it, the location is at L246-L248.

**(9) Reviewer: L 307: please explain the discretization process in more detail**

**Authors:** The authors gave detailed explanations about the discretization rule at Line 223-225 in Section 3.2.

**(10) Reviewer: Section 4: add information of the size of the investigation area could be given earlier.**

**Authors:** we added the size of the investigating area at Line 258 in Section 4.1.

**(11) Reviewer: Figure 9: right: what does the star mean?**

**Authors:** We have added the explanation about the red star at Line 364.

**Reviewer 2**

**(1) Reviewer: the cohesion and friction angle always are not uniform distributed within two limits, the proposed method is not based physically as called by the authors.**

**Authors:** In our paper, the reason that we proposed the physical model is based on the finite slope model. This model is suitable for mechanism analysis for shallow landslide and has been widely used and reported by many researchers, such as the articles written by Iverson et al (1997) and Apip et al (2010). Indeed, the cohesion force (c) and internal friction angle (φ) (the two important parameters in the infinite slope model) is difficult to determine and may not also be of uniform distribution, however, it should be noted that this distribution is only an assumption in our paper and only a method to solve the uncertainty of the two parameters. Additionally, even the two parameters are hardly to determine, they still cannot influence the physical meaning of infinite slope model, so the word of "physics" in the title of this paper is supposed to be retained.

**(2) Reviewer: As expressed in Eq. 1, the safety factor is influenced by phi_b related to the matric suction. How this parameter is determined in the calculation?**

Authors: $\Phi_b$ is the parameter relating to the matrix suction, which is close to the internal friction angle φ in the condition of the low matrix suction. And low matrix suction means high soil water content; this situation is favorable to landslide. So in our paper, $\Phi_b$ is set to be equal to the internal friction angle φ. This explanation is at Line 93.

**(3) Reviewer: How about the initial water content along the depth?**

**Authors:** Initial soil water content was based on the residual water content of each soil type. If the residual water content of a soil type is equal to w, then any pixel belonging to this soil type will be assigned this value. This means that the initial water content of each layer was assumed to be w, namely uniform distribution along the depth. This distribution has some drawbacks, for example, the deeper soil layer may have higher soil water content, but we cannot identify it at the regional scale and have to use the above easy method.

**(4) Reviewer: The distribution of cohesion and internal friction angle in Fig. 8 should be consistent with the soil type in Fig.7, but it is not so now.**

**Authors:** We have explained why Fig.8 is different from Fig.7. Asked by editor, the authors added the detailed process to get these data according to the lithology map and handbook at Line 320-324.

**Editor:**

**(1) Editor:   In addition to reviewer comments, please consider a further discussion in Section 2.3 of the choice of uniform distributions for the sampled parameters, and justification of the sampling scheme connecting sampled values of cohesion force and internal friction angle through the variable r_i.**

**Authors:** The authors added the detailed explanations about the two issues at Line 129-131, and line 139-140.

[revised manuscript text omitted]

---

## Referee Report (RR1)

Consistent with my expertize, I have looked at the design of the MC experiment. I have some suggestions, which the authors may want to implement or recognize.

First, usually a 'raw' MC experiment with no selection will involve some non-physical outcomes, especially when the parameters fall in the corners of the parameter space. With simple models, this might show up as nonsensical model values. If the authors are able to identify these, then it would be best to delete them from the ensemble, and not to include them in either the numerator or the denominator of (6).

Second, the interpretation of the MC experiment is 'integrating out' uncertainty in the parameters. Formally, with theta = (c, phi),

Pr(F_s < 1) = int_theta Pr(F_s < 1, pars = theta) dtheta
            = int_theta Pr(F_s < 1 | pars = theta) Pr(pars = theta) dtheta

where the model gives the first term in the integrand (either a 0 or a 1 in practice) and their choice for the distribution of the parameters gives the second. So when they assign a distribution to the sampling of (c, phi), the authors are in fact describing Pr(c, phi), their 'prior beliefs' about (c, phi). In this context, uniform between c_origin/2 and 2*c_origin is slightly unusual, because the two limits suggest that uncertainty is multiplicative. Better might be U uniform in [-1, 1] and then c = 2^U * c_origin.

Finally, there are only two uncertain parameters here, which means that MC integration is not required. It would be more accurate to use quadrature to integrate over (c, phi).

It is important that the authors and their readers understand that they are computing their probability by marginalizing over the uncertain parameters in a probabilistic model, and that the MC experiment is just a technique to estimate the integration (based, of course, on the weak law of large numbers). Better techniques are available in 2 dimenions.

Jonathan Rougier
Bristol, Jan 2018

---

## Author Response (AR2)

[revised manuscript text omitted]

2008), e.g. the common used functions of normal distribution and the uniform distribution (Schmidt et al., 2008;
Raia et al., 2014). The physical parameters submit the normal distribution meaning that they can be expressed as
$c=N(\mu_c, \sigma_c^2)$, $\varphi=N(\mu_\varphi, \sigma_\varphi^2)$. In this distribution function, $\mu$ represents the mean value of the soil parameters, and
$\sigma$ represents the standard deviation. So if the normal distribution function is adopted to describe the uncertainty,
the two key parameters (mean value $\mu$ and standard deviation $\sigma$) should be firstly determined in order to establish
the corresponding specific distribution function for each pixel within study area. To achieve this purpose, numer-
ous samples and experimental works are necessary and it is very difficult to be implemented in a large region.
Because the uniform distribution suited in the investigation of large areas where information on the
geo-hydrological properties is limited (Raia et al., 2014), which can easily allow authors to get random parameters
from its set approximate variation range instead of large amount of field and experimental works in large area.
Accordingly, the uncertainties of cohesion force and internal friction angle are described here as uniform probabil-
ity distributions in the intervals of $c=U(c_{min}, c_{max})$, and $\varphi=U(\varphi_{min}, \varphi_{max})$, respectively. Then, Monte Carlo method
can be used to randomly extract cohesion force and internal friction angles from the two intervals $n$ times in any
forecasting step. This random approach is used to account for the uncertain nature of soil mechanical parameters.
The detailed description of random extracting process is as follows: the extraction of the two parameters is de-
pendent on the variable $r_i$ which is described as uniform probability distributions in the interval of $r_i=U(0,1)$, the
random values of cohesion force $c_i$ and internal friction angle $\varphi_i$ can be identified via Eq. 3 and Eq.4. In these
equations, $r_i$ can help to get a random number $c_i$ with uniform distribution rule between $c_{min}$ and $c_{max}$, because the
variable $r_i$ submits this distribution rule between 0 and 1. In the whole extracting process, each $r_i$ may have dif-
ferent value and corresponds to a kind of uncertainty of mechanical parameters, but in one extracting step, the
calculated $c_i$ and $\varphi_i$ in Eq. 3 and Eq.4 use a same value of $r_i$.

$$c_i=r_i(c_{max}-c_{min})+c_{min} \tag{3}$$

$$\varphi_i=r_i(\varphi_{max}-\varphi_{min})+\varphi_{min} \tag{4}$$

There, $i$ is the number of some pixel. $c_{min}$ and $\varphi_{min}$ are lower borders of intervals of the two mechanical parameters
expected values; $c_{max}$ and $\varphi_{max}$ are the upper borders. Both the lower and upper borders will vary from pixel to
pixel, because each pixel with different lithology has different mechanical parameters. For any pixel in any fore-

[revised manuscript text omitted]
: (1) $H_w$ representing the instable soil depth in Eq.1 is not equal to the soil depth $L$ in Section 3.2, and cannot be identified in advance. We have to divide each pixel with a certain soil depth $L$ into several soil layers in order to calculate the $Fs$ using Eq.1 layer by layer. When the calculated soil layer is the $j^{th}$, the parameters $H_w$ will be equal to the sum of all the soil layers above the $j^{th}$ layer (including the depth of the $j^{th}$ soil layer). As mentioned in Section 3.2, each pixel was divided into soil layer with a same depth. The matrix suction and soil water content are the important hydrological parameters to the stability analysis of pixel which will be calculated and saved in each divided soil layer after the hydrological process simulation. So we adopt the same discretization rule during the stability analysis in order to easily extract these hydrological parameters(2) The Monte Carlo method is used to extract the cohesion force and the internal friction angle $n$ times from the corresponding intervals ($c=U(c_{min}, c_{max})$, and $\varphi=U(\varphi_{min}, \varphi_{max})$) of each pixel; (3) The safety factor $Fs$ of each divided layer within one pixel is calculated after each extraction, using the soil mechanical parameters and the hydrological parameters only related to time as inputs of Eq.1, when the $F_s$ of $j^{th}$ layer is less than 1, then the calculation process within the pixel will stop; (4) Once the Monte Carlo process end, the total times $Sum_{Fs<1}$  (a count of the number of occurrences satisfying the instability condition) is obtained, and the ratio $P$ of $Fs$ <1 
[revised manuscript text omitted]

**Response to Editor**

Dear Editor,

Thanks a lot for your kind comments on our manuscript, the proposed advices are very helpful to improve our manuscript. Now we have amended this manuscript according to the advice and used the track changes mode in MS to highlight modifications in this manuscript. Any change was also marked by yellow color. The authors will give detailed explanations one by one as follows:

**(1) Editor: Stable conditions:** At line 49-50, the conditions given for stability and instability are not consistent with the definition of Fs elsewhere in the paper. Please ensure that a consistent and correct definition is used throughout the paper.

1. "49: The safety factor of each pixel within a forecasting region, Fs (Fs=R/S: where R is shear resistance and S is the driving force) is calculated considering rainfall infiltration, pixels are then identified as unstable (Fs > 1) or stable (Fs < 1)."

2. 105: From a deterministic point of view, this physical framework can be briefly drawn as follows: for each pixel in the forecast area, if Fs ?? 1 it's considered unstable, while pixels with Fs>1 are considered to be stable.

3. At eqn (6), could the authors please clarify if Sum_Fs<1 is a count of the number of occurrences satisfying the instability condition, or a summation?

4. Line 49-50, Eqn (6), Line 249 and abstract: Please clarify whether the failure condition is Fs < 1 or Fs <= 1, and correct the text and equation 6 accordingly.

**Authors:** The authors appreciated the editor for pointing out this mistake. We have amended these mistakes in our manuscript in order to make them consistent with each other. For example:

1. 49: We changed the original sentence to "pixels are then identified as unstable (Fs > 1) or stable (Fs < 1)" to "pixels are then identified as unstable (Fs < 1) or stable (Fs ≥ 1)" in Line 50.

2. 105: We have modified this sentence in the current form: if $Fs$ < 1 it's considered unstable, while pixels with $Fs$≥ 1 are considered to be stable in Line 106.

3. $Sum_{Fs<1}$ is a count of the number of occurrences satisfying the instability condition. We have added the explanation in Line 266-277.

4. The authors clarify that the failure condition is Fs < 1. The pixel is considered to be stable when Fs ≥ 1.

**(2) Editor: Vertical discretization of the soil pixel**: To address the referee's comments relating to the discretization, please revise the text again, focusing on a more detailed description of the pixel-level forecasting algorithm proposed in Section 3.3. This should:

a) Introduce and justify the choice of 10 vertical levels, commenting on the implications of this scheme with respect to regional variations in soil depth.

b) Describe how Eqn (2) was applied within step (2) of the algorithm, making explicit the mathematical connection between H_s in Eqn (2) and the soil depth within any soil layer j.

**Authors:** The authors appreciate the editor for giving the above excellent advices to the vertical discretization of the soil pixel. The authors added the detailed explanations in the suggested Section 3.3 in our manuscript.

**(3) Editor:** The authors have expanded on the discussion of the probabilistic basis of the model, as requested by the referees and editor. However, the revised text does not fully address the issues raised in the discussion. Two issues that require further attention are:

(a) Referee #2 commented that "the cohesion and friction angle always are not uniform distributed within two limits", which the authors acknowledge may be the case in their response. However, the implications of the uniformity assumption are not explored in the paper nor in the interactive discussion. The new text at lines 129-131 appears to offer a justification of this choice (of uniform distribution) on the grounds that it was easy to implement. However, other distributional assumptions could also be implemented without great difficulty (e.g. a normal or triangular sampling distribution).

The authors are therefore requested to add further discussion of the distributional assumptions made in the paper, so that the robustness of this choice can be assessed.

(b) The new text at lines 139-140 does not address the editor's previous question about the role of the simulated variable r_i. Equations 3 and 4 imply that values drawn for c and phi in each sample are both dependent on a value, r_i, sampled from U(0,1). Please improve the notation to clarify whether r_i in Equation 3 and r_i in Equation 4 represent two independent samples from the U(0,1) distribution. Please also make explicit where subscript i refers to the pixel index, and clarify if c_min, c_max, phi_min and phi_max also vary from pixel to pixel.

**Authors:** we have added some further discussions and explanations in our manuscript. The first modification part is in Line 128-135 in order to further discuss the distributional assumptions made in the paper. The second modification part is in Line 147-154 in order to response the address the editor's previous question about the role of the simulated variable $r_i$, the authors have gave explicit expansions at the corresponding position in our manuscript based on the advices proposed by Editor.

---

## Author Response (AR3)

[revised manuscript text omitted]

used functions of normal distribution and the uniform distribution (Schmidt et al., 2008; Raia et al., 2014). The
physical parameters submit the normal distribution meaning that they can be expressed as $c=N(\mu_c, \sigma_c^2)$, $\varphi=N(\mu_\varphi,$
$\sigma_\varphi^2)$. In this distribution function, $\mu$ represents the mean value of the soil parameters, and $\sigma$ represents the standard
deviation. So if the normal distribution function is adopted to describe the uncertainty, the two key parameters
(mean value $\mu$ and standard deviation $\sigma$) should be firstly determined in order to establish the corresponding spe-
cific distribution function for each pixel within study area. To achieve this purpose, numerous samples and ex-
perimental works are necessary and it is very difficult to be implemented in a large region. Because the uniform
distribution suited in the investigation of large areas where information on the geo-hydrological properties is lim-
ited (Raia et al., 2014), which can easily allow authors to get random parameters from its set approximate varia-
tion range instead of large amount of field and experimental works in large area. Accordingly, the uncertainties of
cohesion force and internal friction angle are described here as uniform probability distributions in the intervals of
$c=U(c_{min}, c_{max})$, and $\varphi=U(\varphi_{min}, \varphi_{max})$, respectively. Then, Monte Carlo method can be used to randomly extract
cohesion force and internal friction angles from the two intervals $n$ times in any forecasting step. This random
approach is used to account for the uncertain nature of soil mechanical parameters. The detailed description of
random extracting process is as follows: the extraction of the two parameters is dependent on the variables $r_{ic}$ and
$r_{i\varphi}$  are described as uniform probability distributions in the interval of $r_{ic}=U(0,1)$, $r_{i\varphi}=U(0,1)$, the ran-
dom values of cohesion force $c_i$ and internal friction angle $\varphi_i$ can be identified via Eq. 3 and Eq.4.
The pa-
$r_{ic}$ and $r_{i\varphi}$ used for calculating $c_i$ and $\varphi_i$ in
Eq. 3 and Eq.4 may have different values because they are independently extracted from (0,1).

$$c_i = r_{ic}(c_{max}-c_{min})+c_{min} \qquad (3)$$

$$\varphi_i = r_{i\varphi}(\varphi_{max}-\varphi_{min})+\varphi_{min} \qquad (4)$$

There, $i$ is the number of some pixel, $c_{min}$ and $\varphi_{min}$ are lower borders of intervals of the two mechanical parameters
expected values; $c_{max}$ and $\varphi_{max}$ are the upper borders. Both the lower and upper borders will vary from pixel to
pixel, because each pixel with different lithology has different mechanical parameters. For any pixel in any fore-
casting step, a matrix $M_i$ can be generated after the $n$-times random extraction process:

$$M_i = [c_i, \varphi_1] = \begin{bmatrix} c_1 & \varphi_1 \\ c_2 & \varphi_2 \\ c_3 & \varphi_3 \\ \cdots & \cdots \\ c_n & \varphi_n \end{bmatrix} \tag{5}$$

[revised manuscript text omitted]

From Eq.3 and Eq.4, it is necessary to identify the lower and upper border of intervals of the soil mechanical
parameters. However, the exact values for lower ($c_{min}$ and $\varphi_{min}$) and upper ($c_{max}$ and $\varphi_{max}$) limits are very difficult
to determine. From currently published papers, there is no known theoretical or experimental method to solve this
issue. Raia et al. (2014) used variations of 1%, 10% and 100% around the values of cohesion force and internal
friction angle (from field tests) to get several intervals, showing that the forecasting effectiveness is significantly
improved by using a large variations. Consequently, this method applies a variation of 100% around the mean
value of these parameters for each pixel to set the corresponding lower and upper borders as follows:

$$c_{\mathrm{random}} \in [0.5 \times c_{\mathrm{origin}}, 2 \times c_{\mathrm{origin}}] \tag{7}$$

$$\varphi_{\mathrm{random}} \in [0.5 \times \varphi_{\mathrm{origin}}, 2 \times \varphi_{\mathrm{origin}}] \tag{8}$$

Where $c_{\mathrm{random}}$ and $\varphi_{\mathrm{random}}$ are the randomly extracted cohesion forces and internal friction angles, $c_{\mathrm{origin}}$ and $\varphi_{\mathrm{origin}}$
are the mean value of each pixel (in this case from the rock mechanics handbook (Ye et al., 1991)).

**3.2 Pixel level hydrological process simulation**

The simulation of hydrological processes including rainfall interception, infiltration, and evapotranspiration is
extremely complicate. However, rainfall infiltration is the key factor in the distribution of soil water content in
underlying surface which simplify the analysis. In southwestern region of China slopes are almost unsaturated
before the rainy season due to characteristic distribution of rainfall influenced by monsoon (Zhang et al., 2014b).
The infiltration process in the vertical direction in unsaturated soil mass can be described by the 1D Richards's
equation (1931):

$$\frac{\partial \theta}{\partial t} = \frac{\partial}{\partial z} \left[ D(\theta) \frac{\partial \theta}{\partial z} \right] - \frac{\partial K(\theta)}{\partial \theta} \tag{9}$$

Where $\theta$ is soil water content, $D(\theta)=K(\theta)/(\mathrm{d}\theta/\mathrm{d}\psi)$ is the hydraulic diffusivity, $\psi$ is the suction of unsaturated soil, $z$
represents the soil depth, which is positive along the soil depth and have the topsoil as the origin point, $K(\theta)$ is the
hydraulic conductivity. The matrix suction is the dominant external force to drive the water movement in unsatu-
rated soil mass, which can be calculated from Eq. 2.

Infiltration upper border: If the topsoil is unsaturated, it has a strong infiltration capacity (Lei et al., 1988).
Then, while the rainfall intensity is less than the infiltration capacity of the topsoil, all precipitation will infiltrate
into topsoil without any runoff. In this scenario, the infiltration border is governed by Eq. (10):

$$-D(\theta) \frac{\partial \theta}{\partial z} + K(\theta) = R(t), \quad t > 0, z = 0 \tag{10}$$

Where $R(t)$ is the rainfall intensity at time $t$. Here, the part of precipitation that exceeds the capacity of infiltration
of the topsoil will transform into runoff (no water storage above topsoil). In this case the topsoil of a pixel is con-
sidered saturated. Thus, the Eq.10 that governs infiltration upper border is transformed into the equation of $\theta=\theta_s$
(Lei et al., 1988). There $\theta_s$ is the saturated moisture corresponding to the soil type.

Infiltration bottom border: It has been experimentally demonstrated that the soil water content beyond a soil
depth of 40 cm is barely influenced by rainfall infiltration (Cui et al., 2003). Consequently a region with a
groundwater level near the surface of the soil has hydrological characteristics in which rainfall infiltration can
hardly induce any groundwater level variation. In this case, it is reasonable to ignore the water exchange process
between the lower boundary and groundwater (Zhang et al., 2015).

An implicit finite difference method is used for discretization of the 1D differential equation of water movement. The calculation time $t$ is segmented into several intervals with the same time gap $\triangle t$, and the soil depth $L$ of each pixel is segmented into soil layers (each layer is named of $i$ number) with the same depth $\triangle z$.

Identifying the initial soil water content is an important issue during the hydrological simulation process. However, this value cannot be directly determined at any given time for a large region due to complex rainfall infiltration and evapotranspiration interactions. In the case of southwestern China, the winter is generally a relatively dry season, thus the soil water content value of the topsoil is very low closing to the residual water content of the soil type (Zhang et al., 2014b). This situation is exploited setting the simulation time to start on January 1 of the forecasting year (driest month in winter), which allows the use of the residual water content corresponding to the soil type as and the initial value of the topsoil water content. Measured meteorological data from January 1 are then feed to the simulation, which allows for a relatively accurate initial value of soil water content for the landslide forecasting. Each simulation step takes also into account the rainfall interception and evapotranspiration processes by means of the algorithm of distributed hydrological model GBHM (Yang et al., 2002).

After the hydrological simulation process identify the initial soil water content of each pixel, the simulation focuses on the extraction of key hydrological parameters (soil water content and matrix suction) necessary for the stability calculation of each pixel using the expected rainfall from Doppler radar forecasting. During this last stage in the simulation in which landslide forecasting is performed, the evapotranspiration processes is not considered since this period is typically short, with rainfalls, negligible sunshine and lower temperatures.

**3.3 Probabilistic landslide forecasting at pixel level**

During the forecasting stage, the hydrological parameters (soil water content and matrix suction) of each pixel in each forecasting step $\Delta t$ are extracted via hydrological process simulation. Then the ratio $P$ is computed for each pixel in several steps as follows: (1) $H_s$ representing the instable soil depth in Eq.1 is not equal to the soil depth $L$ in Section 3.2, and cannot be identified in advance. We have to divide each pixel with a certain soil depth $L$ into several soil layers in order to calculate the $Fs$ using Eq.1 layer by layer. When the calculated soil layer is the $j^{th}$, the parameters $H_s$ will be equal to the sum of all the soil layers above the $j^{th}$ layer (including the depth of the $j^{th}$ soil layer). As mentioned in Section 3.2, each pixel was divided into soil layer with a same depth. The matrix suction and soil water content are the important hydrological parameters to the stability analysis of pixel which will be calculated and saved in each divided soil layer after the hydrological process simulation. So we adopt the same discretization rule during the stability analysis in order to easily extract these hydrological parameters(2) The Monte Carlo method is used to extract the cohesion force and the internal friction angle $n$ times from the corresponding intervals ($c=U(c_{min}, c_{max})$, and $\varphi=U(\varphi_{min}, \varphi_{max})$) of each pixel; (3) The safety factor $Fs$ of each divided layer within one pixel is calculated after each extraction, using the soil mechanical parameters and the hydrological parameters only related to time as inputs of Eq.1, when the $F_s$ of $j^{th}$ layer is less than 1, then the calculation process within the pixel will stop; (4) Once the Monte Carlo process end, the total times $Sum_{Fs<1}$ (a count of the number of occurrences satisfying the instability condition) is obtained, and the ratio $P$ of $Fs <1$ 
[revised manuscript text omitted]

**Response to Editor**

Dear editor:

Thanks a lot for your kind comments on our manuscript, the proposed advices are very helpful to improve our manuscript. Now we have amended this manuscript according to the advice and used the track changes mode in MS to highlight modifications in this manuscript. Any change was also marked by yellow color. The authors will give detailed explanations one by one as follows:

On Line 144-153 in Page 4: after careful consideration, the authors think it is not proper to impose the correlation of cohesion force and internal friction angle. So we have accepted the advices and changed the main expressions and highlighted the independent attributions of the two parameters.

On lines 356-360 in Page 12-13, and on Line 427-431 in Page 15-16: the authors have renewed the calculation results according to the independently extracting method. As shown in the renewed analysis results in Page 12-13 and Page 427-431, however, the independent extracting method has slightly affection on the results. Probably, the reason is that many parameters (such as the suction and soil water content) can affect the safety factors $Fs$. Although the new independent extracting method of the $r_{ic}$ and $r_{i\varphi}$ is different from the original parameter $r_i$, they still belong to the interval (0, 1) and probably did not have significant influence on the cohesion force ($c$) and the internal friction angle ($\varphi$).

The authors think the editor and reviewer have proposed a very interesting topic about the correlation of the cohesion force and internal friction angle of the soil. In these days, the authors have indexed numerous references, but did not find any paper about the correlation of the two parameters. The authors will study this topic via testing soil samples in the future.